# A validation of machine learning-based risk scores in the prehospital setting

Douglas Spangler[1]*, Thomas Hermansson[2], David Smekal[1,2], Hans Blomberg[1,2]

**1** Uppsala Center for Prehospital Research, Department of Surgical Sciences—Anesthesia and Intensive Care, Uppsala University, Uppsala, Sweden, **2** Uppsala Ambulance Service, Uppsala University Hospital, Uppsala, Sweden

* douglas.spangler@akademiska.se

## Abstract

### Background

The triage of patients in prehospital care is a difficult task, and improved risk assessment tools are needed both at the dispatch center and on the ambulance to differentiate between low- and high-risk patients. This study validates a machine learning-based approach to generating risk scores based on hospital outcomes using routinely collected prehospital data.

### Methods

Dispatch, ambulance, and hospital data were collected in one Swedish region from 2016–2017. Dispatch center and ambulance records were used to develop gradient boosting models predicting hospital admission, critical care (defined as admission to an intensive care unit or in-hospital mortality), and two-day mortality. Composite risk scores were generated based on the models and compared to National Early Warning Scores (NEWS) and actual dispatched priorities in a prospectively gathered dataset from 2018.

### Results

A total of 38203 patients were included from 2016–2018. Concordance indexes (or areas under the receiver operating characteristics curve) for dispatched priorities ranged from 0.51–0.66, while those for NEWS ranged from 0.66–0.85. Concordance ranged from 0.70–0.79 for risk scores based only on dispatch data, and 0.79–0.89 for risk scores including ambulance data. Dispatch data-based risk scores consistently outperformed dispatched priorities in predicting hospital outcomes, while models including ambulance data also consistently outperformed NEWS. Model performance in the prospective test dataset was similar to that found using cross-validation, and calibration was comparable to that of NEWS.

### Conclusions

Machine learning-based risk scores outperformed a widely-used rule-based triage algorithm and human prioritization decisions in predicting hospital outcomes. Performance was robust in a prospectively gathered dataset, and scores demonstrated adequate calibration. Future research should explore the robustness of these methods when applied to other settings,

**Data Availability Statement:** Our ethics approval limits us to the publication of results at the aggregate level only, precluding us from publishing individual-level patient data. The Swedish Data Protection Authority has furthermore not yet

endorsed a process for the anonymization of individually identifiable data which could be applied to ensure compliance with the EU General Data Protection Regulation in publishing this type of sensitive data. We include all source code necessary to replicate the reported results in other settings as S1 Code, along with a randomized synthetic data set. Data underlying the results are owned by the Uppsala Ambulance Service, and are available for researchers who meet the criteria for access to confidential data. Please contact the Uppsala Ambulance Service at ambulanssjukvard@akademiska.se to arrange access to the data underlying this study.

**Funding:** HB received funding for this study from the Swedish Innovation Agency (https://www. vinnova.se, grant number 2017-04652). The funders had no role in study design, data collection and analysis, decision to publish, or preparation of the manuscript.

**Competing interests:** The authors have declared that no competing interests exist.

establish appropriate outcome measures for use in determining the need for prehospital care, and investigate the clinical impact of interventions based on these methods.

## Introduction

Emergency care systems in the developed world face increasing burdens due to an aging population [1–4], and in prehospital care it is often necessary to prioritize high-risk patients in situations where resources are scarce. Prehospital care systems have also increasingly sought to identify patients not in need of emergency care, and to direct these patients to appropriate forms of alternative care both upon contact via telephone with the dispatch center, and upon the arrival of an ambulance to a patient [5–12]. Performing these tasks safely and efficiently requires not only well trained prehospital care providers and carefully considered clinical guidelines, but also the employment of triage algorithms able to perform risk differentiation across the diverse cohort of patients presenting to prehospital care systems.

Systems to differentiate high- and low-risk patients in prehospital care have traditionally relied on simple rule-based algorithms. Many commonly used algorithms seek to identify specific high-acuity conditions within certain subsets of patients such as cardiac arrest, trauma, or stroke [13–15]. Other algorithms are intended for use within a broader cohort of patients, including the vital-sign based Critical Illness Prediction (CIP) and National Early Warning Scores (NEWS) [16–20], and the Medical Priority Dispatching System (MPDS) [21] for triage performed over the telephone. In applying such tools, providers commonly "over-triage" patients, as false negatives are thought to be associated with far greater costs than false positive findings [22–25]. In this study, we focus on NEWS due to its widespread international use in the context of prehospital care. NEWS is intended for use as a tool to identify patients at risk for deterioration, and in studies validating NEWS, patient deterioration has been operationalized in several ways, primarily focusing on patient mortality and/or admission to an Intensive Care Unit (ICU) [20].

In the context of Emergency Department (ED) triage, Machine Learning (ML) based triage algorithms have been shown to out-perform their rule-based counterparts in predicting patient outcomes including mortality, ICU care, and hospital admission [26–29]. We identified no research relating to the ability of prehospital data to similarly predict hospital outcomes, though there are indications that ML techniques may be effective in identifying specific high-acuity conditions such as cardiac arrest at the dispatch center [30]. ML-based approaches offer the potential to analyze large and complex sets of predictors, and automatically calculate risk scores for use by care providers. While ML methods can provide substantial gains in terms of accuracy over traditional risk assessment tools, they also have drawbacks. Perhaps most concerningly, ML methods readily integrate undesirable systemic biases present in the data they are trained on into the prediction model [31].

In this study we propose the use of a composite risk score representing the average probability of several relevant outcomes occurring. This differs from the approaches of previous researchers, who have either investigated only single measures of patient outcome [27,28], or binned model predictions across specific ranges of predicted likelihoods [26,32]. To generate composite scores, we trained separate models for each outcome and to then averaged their predictions. In averaging multiple model predictions, we argue that it becomes decreasingly likely that the same sources of undesirable bias (e.g., idiosyncrasies in hospital admission practices, or differences in the ability of care providers to prevent mortality from different causes) will

exist across all models. Thus, the combined predictions as embedded in the composite risk score is argued to be a less biased representation of the true underlying risk for deterioration associated with a patient than any single measure.

There are several potential use cases for such ML-based risk scores in prehospital care. At the dispatch center, scores could be used to improve the ability to prioritize patients in resource constrained situations, enabling faster ambulance responses for higher-risk patients and the referral of low risk patients to less resource intensive forms of care. Similarly, it may be possible to develop more accurate risk assessment instruments for use by ambulance staff. Given sufficiently high levels of sensitivity, such instruments could be used by care providers to identify low-risk patients suitable for referral to primary care services, thus alleviating the increasingly vexing problem of overcrowding at EDs [33–35]. At higher levels of specificity, instruments could be used to automatically alert receiving EDs to incoming high-risk patients, and allow earlier activation of hospital care processes.

To accommodate this breadth of potential use cases, this study aims to validate our proposed methods empirically in a broad cohort of patients based on the data available at two distinct points in the chain of emergency care: Over the telephone at the Emergency Medical Dispatch (EMD) center, and after an ambulance has made contact with the patient in the field. We investigated the feasibility of using these methods to improve the decisional capacity prehospital care providers in these settings by comparing their accuracy with a previously validated and widely used triage algorithm (NEWS), and with prioritization decisions made by nurses at the EMD center per current clinical practice.

## Methods

### Source of data

This study took place in the region of Uppsala, Sweden, with a size of 8 209 km2, and a population of 376 354 in 2018. The region is served by two hospital-based EDs, a single regional EMD center staffed by Registered Nurses (RNs) employing a self-developed Clinical Decision Support System (CDSS), and 18 RN-staffed ambulances. The CDSS consists of an interface wherein dispatchers first seek to identify a set life-threatening conditions (cardiac/respiratory arrest or unconsciousness), and then document the primary complaint of the patient. Based on the documented complaint, a battery of questions is presented, the answers to which determine the priority of the call, or open additional complaints. While the specific set of questions are idiosyncratic to this and 3 other Swedish regions, its structure is similar to other dispatch CDSS such as the widely-used MPDS [21].

Ambulance responses are triaged by an RN to one of four priority levels, with 1A representing the highest priority calls (e.g. cardiac/respiratory arrest), and 1B representing less emergent calls still receiving a "lights and sirens" (L&S) response. Calls with a priority of 2A represent urgent, but non-emergent ambulance responses, while 2B calls may be held to ensure resource availability.

Records from January 2016 to December 2017 were extracted to serve as the basis for all model development. Upon finalizing the methods to be reported upon, records from January to December 2018 were extracted to form a test dataset to investigate the prospective performance of the models. The data in this study were extracted from databases owned by the Uppsala ambulance service containing dispatch, ambulance, and hospital outcome data collected routinely for quality assurance and improvement purposes. Ambulance records were deterministically linked to dispatch records based on unique record identifiers available in both systems. Hospital records were extracted from the regional Electronic Medical Records (EMR) system based on patient Personal Identification Numbers (PINs) collected either by dispatchers

or ambulance crews. This study was approved by the Uppsala regional ethics review board (dnr 2018/133), which waived the requirement for informed consent. Identifiable data was handled by hospital employees only, and all research was performed on de-identified datasets.

## Participants

All dispatch records associated with a primary ambulance response to a single-patient incident (i.e., excluding multi-patient traffic accidents and planned inter-facility transports) were selected for inclusion. Records lacking documentation in the CDSS used at the EMD center were excluded, as were records in which an invalid PIN or multiple PINs were documented. Dispatch records with no associated ambulance journal (e.g. calls cancelled *en route*, or where no patient was found), and records indicating that the patient was treated and left at the scene of the incident were excluded. We further excluded records where no EMR system entry associated with the patient at the appropriate time could be identified (typically due to documentation errors, or transports to facilities outside of the studied region), and EMR system records indicating that the patient was transported to a non-ED destination (e.g. a primary/urgent care facility, or a direct admission to a hospital ward). We also excluded patients with ambulance records missing measurements of more than two of the vital signs necessary to calculate a NEWS score. Patients under the age of 18 were excluded as NEWS scores are not valid predictors of risk for pediatric patients.

## Outcomes

We selected three outcome measures based on their face validity in representing a range of outcome acuity levels, and based to their use in previous studies; 1) patient admission to a hospital ward [26–28,32], 2) the provision of critical care, defined as admission to an Intensive Care Unit (ICU) or in-hospital mortality [26,28], and 3) all-cause patient mortality within two days [18,19]. In consideration of the need to update models continuously upon implementation, we included only in-hospital deaths occurring within 30 days of contact with the ED in calculating the critical care measure.

We generated composite risk scores by combining independent model predictions for each of these outcomes. The method we propose results in composite risk scores reflecting the normalized mean likelihood of several outcomes with face validity as being representative of patient acuity occurring, without incurring the loss of information associated with binning continuous variables. We applied no weights in the compositing process, as the relative importance of these measures in in establishing the overall acuity of the patient is not known.

## Predictors

Predictors extracted from the dispatch system included patient demographics (age and gender), the operational characteristics of the call (Hour and month that the call was received, haversine distance to the nearest ED, and prior contacts with the EMD center by the patient), and the clinical characteristics of the call as documented in the existing rule-based CDSS. We included the 59 complaint categories, and the 1592 distinct question and answer combinations available in the CDSS as potential predictors in our models. Each of the questions in the CDSS was encoded with a 1 representing a positive answer to the question, and 0 representing a negative answer to the question. Questions with multiple potential answers were encoded on a numerical scale in cases where the answers were ordinal (e.g., "How long have the symptoms lasted?"), and as dummy variables if the answers were non-ordered. The recommended priority of the call based on the existing rule-based triage system was also included as a predictor in the dispatch dataset.

Predictors extracted from ambulance records represented the information which would be available at the time of patient hand-over to ED staff, and included the primary and secondary complaints, additional operational characteristics (times to reach the incident, on scene, and to the hospital), vital signs, patient history, medications and procedures administered, and the clinical findings of ambulance staff. Descriptive statistics for the included predictors are reported in S1 Table.

To provide a basis for comparison, we extracted the dispatched priority of the call as determined by the RN handling the call at the EMD center, and retrospectively calculated NEWS scores for each included patient. If multiple vital sign measurements were taken, the first set was used both as model predictors and to calculate risk scores based on the NEWS 2 algorithm (42) as commonly done in other validation studies (20).

## Missing data

Missing vital sign measurements in ambulance records are not likely to be missing completely at random, and must be considered carefully [36,37]. Based on exploratory analysis and clinical judgement, we surmised that records missing at most two of the vital signs constituting the NEWS score fulfilled the missing at random assumption necessary to perform multiple imputation. Missing vitals were multiply imputed five times using predictive mean matching over 20 iterations as implemented in the 'mice' R package [38]. The characteristics of the imputed data were examined, and the median of the imputed vital signs was used to calculate NEWS scores. Multiply imputed data were not used as predictors, with missing data handled natively by the ML models used here.

## Statistical analysis

We entered each set of predictors transformed as previously described into gradient boosting models as implemented in the XGBoost R package [39]. This algorithm involves the sequential estimation of multiple weak decision trees, with each additional tree reducing the error associated with the previously estimated trees [40]. Model predictions were combined into composite risk scores by scaling each set of outcome predictions to have a population mean of zero and a standard deviation of one. A log transformation applied to improve calibration and interpretability. All component values were then averaged, resulting in a composite risk score following an approximately standard normal distribution.

We investigated model discrimination based primarily on Receiver Operating Characteristics (ROC) curves, using the area under these curves (a measure equivalent to the concordance index, or c-index of the model) as summary performance measures [40]. Precision/Recall curves and their corresponding areas under the curve are included in S1 Analysis. 95% confidence intervals for descriptive statistics and c-index values were generated based on the percentiles of 1000 basic bootstrap samples (using stratified resampling for c-index values) as implemented in the 'boot' R package [41]. Model calibration overall and in a number of subpopulations was investigated visually using lowess smoothed calibration curves, and summarized using the mean absolute error between predicted and ideally calibrated probabilities using the 'val.prob' function from the 'rms' R package [42]. As a secondary analysis, we considered the hypothetical situation in which calls had been assigned priority levels at the dispatch center based on the dispatch data-based risk scores, with threshold values calibrated to match the distribution of patient volume identified in our data.

We considered the performance of the models in the prospective dataset to be the best metric of future model performance, though results in this field have previously been reported based on cross-validation [26,32] or randomly selected hold-out samples [28]. In this paper we

report our main findings based on model performance in a prospective test dataset, and include results based on cross-validation for comparison. Model performance in the training dataset was estimated using 5-fold cross-validation (CV), and model performance in the testing dataset was based on models estimated using the full training dataset.

Readers interested in further details of the methods employed to produce the results reported here are encouraged to peruse the commented source code found in S1 Code. All model development and validation was performed using R version 3.5.3 [43].

## Results

### Participants

A total of 68 668 records were collected, of which 45 045 were in the training dataset, and 23 623 were in the test dataset as reported in Table 1. Overall, 30 465 records (44%) were excluded due all criteria. A lower proportion of records were excluded from the test dataset, primarily due to fewer non-matched ambulance and hospital records.

Summary statistics describing the characteristics of all patients included in the study (across both training and testing sets), both in total and stratified by dispatched priority are presented in Table 2. We found that ambulance predictors and outcomes were generally distributed such that higher priority calls had higher levels of patient acuity, with the notable exception of hospital admission which remained constant at around 50% regardless of dispatched priority. Higher priority patients were generally younger, more often male, and had a higher proportion of missing vital signs. Overall, at least one vital sign was missing in a quarter of ambulance records, with the most commonly missing vital sign measurement being the patient's body temperature. Temperature was missing in 15% of cases, and other vital signs were missing in less than 5% of cases as reported in S1 Table. Multiple imputation of these vital signs resulted in good convergence and similarity to non-imputed data, and NEWS scores based on sets of imputed scores did not differ significantly in terms of predictive value.

### Model performance

Receiver operating characteristics curves across the three hospital outcomes for each of the risk prediction scores, as well as for the dispatched priority of the call are presented in Fig 1. We found that for all investigated outcomes, risk scores based on ambulance data outperformed all other instruments investigated. NEWS scores had a greater overall c-index than

**Table 1. Results of applying exclusion criteria.**

|  | Training dataset (2016–2017) | | | Test dataset (2018) | | |
|---|---|---|---|---|---|---|
|  | Excluded, N | Excluded, percent | Remaining, N | Excluded, N | Excluded, percent | Remaining, N |
| Original |  |  | 45045 |  |  | 23623 |
| No dispatch CDSS data | 2358 | 5.5 | 42687 | 857 | 3.8 | 22766 |
| Missing PIN | 2113 | 5.2 | 40574 | 1244 | 5.8 | 21522 |
| No ambulance journal | 2526 | 6.6 | 38048 | 933 | 4.5 | 20589 |
| No ambulance transport | 3879 | 11.4 | 34169 | 2461 | 13.6 | 18128 |
| No hospital journal | 3958 | 13.1 | 30211 | 1429 | 8.6 | 16699 |
| No ED visit | 2939 | 10.8 | 27272 | 1590 | 10.5 | 15109 |
| Missing > 2 vitals | 1336 | 5.2 | 25936 | 829 | 5.8 | 14280 |
| Patient age < 18 | 1328 | 5.4 | 24608 | 685 | 5 | 13595 |
| Final | 20437 | 45.4 | 24608 | 10028 | 42.5 | 13595 |

**Table 2. Descriptive statistics of included population.**

| | Priority | | | | |
| --- | --- | --- | --- | --- | --- |
| | 1A | 1B | 2A | 2B | Total |
| N | 1283 | 15533 | 17227 | 4160 | 38203 |
| Age, mean | 56.2 (54.8–57.5) | 64.5 (64.1–64.8) | 67.5 (67.2–67.8) | 67.3 (66.6–67.9) | 65.9 (65.6–66.1) |
| Female, percent | 46.1 (43.3–48.9) | 49.4 (48.6–50.1) | 53.9 (53.1–54.6) | 54.5 (53.0–56.0) | 51.9 (51.4–52.4) |
| Emergent transport, percent | 38.7 (35.9–41.5) | 24.6 (23.8–25.2) | 4.3 (4.0–4.6) | 2.2 (1.8–2.6) | 13.5 (13.1–13.8) |
| Ambulance intervention[a], percent | 87.9 (86.1–89.6) | 87.4 (86.9–87.9) | 71.1 (70.4–71.8) | 62.1 (60.6–63.6) | 77.3 (76.9–77.7) |
| Missing vitals, percent | 33.8 (31.3–36.5) | 25.7 (25.0–26.4) | 24.4 (23.7–25.1) | 23.8 (22.5–25.2) | 25.2 (24.7–25.6) |
| NEWS value, mean | 5.79 (5.57–6.00) | 3.76 (3.70–3.82) | 2.97 (2.92–3.01) | 2.40 (2.32–2.48) | 3.32 (3.29–3.36) |
| Prior contacts (30 days), mean | 0.21 (0.17–0.24) | 0.17 (0.16–0.18) | 0.17 (0.16–0.18) | 0.23 (0.21–0.24) | 0.18 (0.17–0.18) |
| Intensive Care Unit, percent | 10.0 (8.5–11.6) | 3.5 (3.1–3.7) | 1.7 (1.5–1.9) | 1.6 (1.2–2.0) | 2.7 (2.5–2.8) |
| In-hospital death, percent | 8.7 (7.2–10.2) | 4.0 (3.7–4.3) | 3.7 (3.4–4.0) | 3.9 (3.4–4.5) | 4.0 (3.8–4.2) |
| Critical care, percent | 15.7 (13.8–17.6) | 6.6 (6.2–6.9) | 4.7 (4.4–5.0) | 4.4 (3.8–5.1) | 5.8 (5.6–6.0) |
| Admitted, percent | 51.9 (49.1–54.6) | 52.3 (51.5–53.0) | 52.3 (51.5–53.1) | 49.2 (47.6–50.6) | 52.0 (51.5–52.5) |
| 2-day mortality, percent | 4.8 (3.6–5.9) | 1.6 (1.4–1.8) | 0.7 (0.6–0.9) | 0.7 (0.5–1.0) | 1.2 (1.1–1.3) |

Statistics are reported with their bootstrapped 95% confidence interval

[a]Interventions include Medication administration, Oxygen administration, IV placement, Spinal/longbone immobilization, 12-lead EKG capture/transmission to hospital, Emergent transport (using lights and sirens), Hospital pre-arrival notification, and administration of CPR.

dispatch data-based models for critical care and two-day morality, but at threshold values corresponding to high levels of sensitivity, dispatch data-based risk predictions provided similar levels of specificity. While we identified no guidelines on appropriate levels of sensitivity with regards to these outcomes, in the context of trauma care, the American College of Surgeons Committee on Trauma (ACS-CoT) recommend that decision rules to identify patients suitable for direct transport to a level-1 trauma center to have a sensitivity of 95% [25,44]. A similarly high level of sensitivity is reasonable in applications where risk scores are used to direct patients to alternate destinations. The level of 95% sensitivity is denoted in Fig 1 by a dotted line.

In predicting critical care, NEWS scores were unable to achieve a level of 95% sensitivity, with a decision rule based on a NEWS score of 1 or more yielding a sensitivity (and 95% CI) of 0.92 (0.90–0.93) and a specificity of 0.24 (0.24–0.25). At the same level of sensitivity, the dispatch and ambulance data-based risk scores yielded specificities of 0.25 (0.24–0.26) and 0.35 (0.35–0.36) respectively. With regards to 2-day mortality, a decision rule based on NEWS score of 2 or above yields a sensitivity of 0.95 (0.91–0.98), corresponding to the ACS-CoT recommendation, while providing a specificity of 0.41 (0.40–0.42). At equivalent levels of sensitivity, the dispatch and ambulance based models provide specificities of 0.30 (0.29–0.31) and 0.48 (0.47–0.49) respectively.

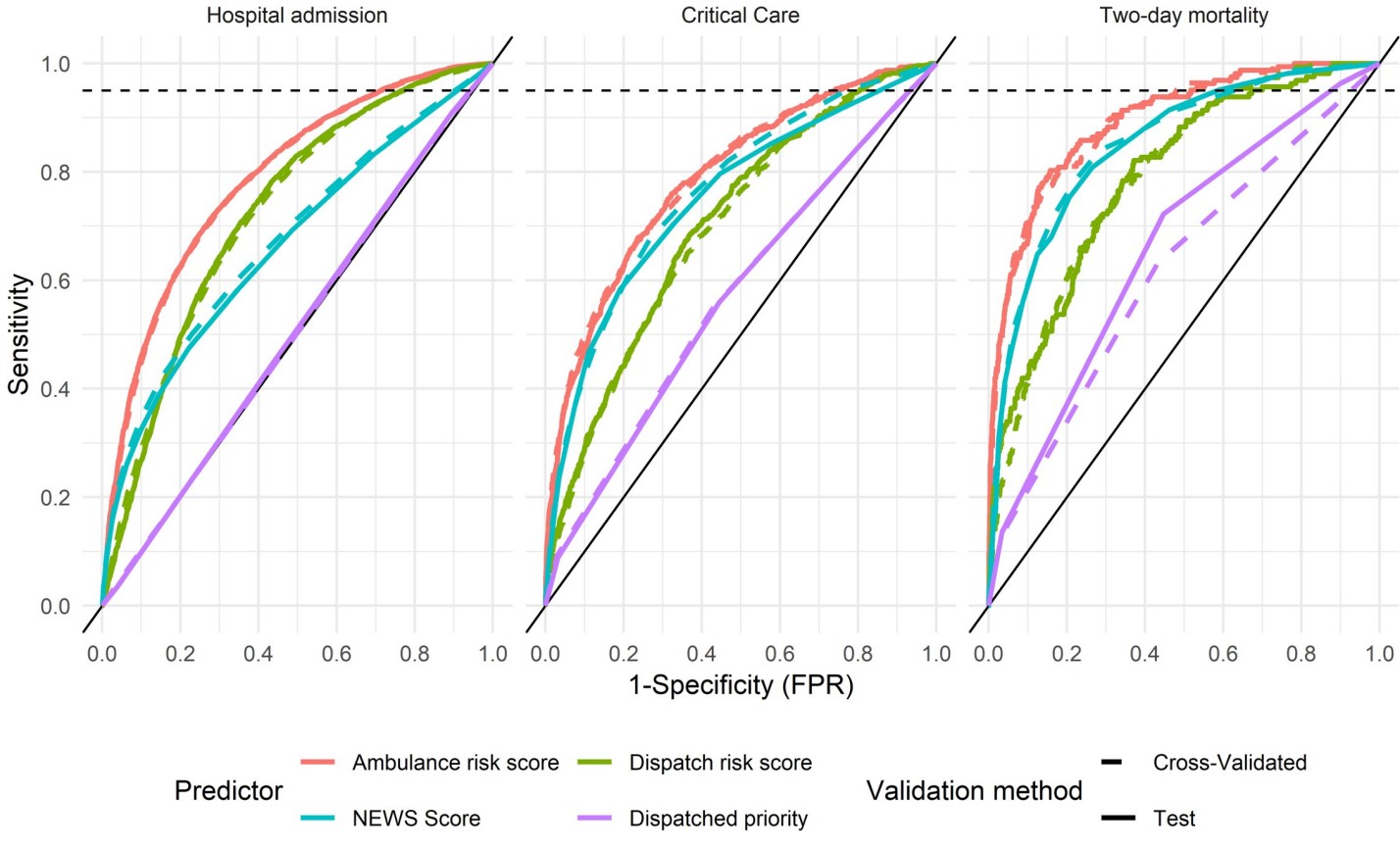

**Fig 1. Receiver operating characteristics in predicting hospital outcomes.** Dotted line corresponds to 95% sensitivity.

Table 3 summarizes the discrimination of the risk assessment instruments for each outcome in the test dataset using the c-index of the model and its 95% confidence interval. ML models based on ambulance data outperformed NEWS scores in terms of c-index for all outcomes. The dispatch data-based risk predictions outperformed NEWS in predicting hospital admission, while NEWS scores outperformed the dispatch data-based predictions for critical

**Table 3. Concordance indexes in predicting hospital outcomes.**

| Validation method | Outcome | Dispatched priority | NEWS Score | Dispatch risk score | Ambulance risk score |
|---|---|---|---|---|---|
| Test | Hospital admission | 0.51 (0.50–0.52) | 0.66 (0.65–0.67) | 0.73 (0.72–0.73) | 0.79 (0.78–0.80) |
| | Critical Care | 0.57 (0.55–0.59) | 0.75 (0.73–0.77) | 0.70 (0.68–0.72) | 0.79 (0.77–0.81) |
| | Two-day mortality | 0.66 (0.62–0.70) | 0.85 (0.81–0.88) | 0.79 (0.76–0.82) | 0.89 (0.87–0.92) |
| Cross-Validated | Hospital admission | 0.50 (0.50–0.51) | 0.67 (0.67–0.68) | 0.72 (0.72–0.73) | 0.79 (0.78–0.79) |
| | Critical Care | 0.57 (0.56–0.59) | 0.76 (0.75–0.78) | 0.70 (0.68–0.71) | 0.79 (0.78–0.80) |
| | Two-day mortality | 0.62 (0.59–0.65) | 0.85 (0.83–0.87) | 0.79 (0.77–0.81) | 0.89 (0.87–0.91) |

C-indexes are reported with their bootstrapped 95% confidence interval

care and two-day mortality in terms of overall discrimination. All risk assessment instruments outperformed dispatched priorities in predicting hospital outcomes, which were found to have some predictive power for critical care and two-day mortality, but none for hospital admission. We found no significant differences between model performance using cross-validation and validation in the test dataset.

We found that both NEWS and ML-based risk scores demonstrated some deviation from ideal calibration as reported in S1 Fig. In terms of mean average error, NEWS scores demonstrated better overall calibration in predicting hospital admission and critical care, but not two-day mortality as reported in S2 Table. In investigating model calibration in sub-populations stratified by age, gender, dispatched priority and patient complaint, some sub-populations did deviate from ideal calibration among both NEWS scores and ML risk scores, though deviations were not consistent across outcomes.

We found that prioritizations at the dispatch center based solely on the dispatch data model would have resulted in substantial improvements in risk differentiation with regards to hospital outcomes and NEWS scores, as reported in Table 4. The proportion of the highest priority patients receiving critical care for instance would increase from the current level of 15% to 22%, while the corresponding levels for the lowest priority patients would decrease from 4% to 1%. With regards to the pre-hospital interventions we included however, differentiation was not improved and indeed in most cases was poorer.

The relative gain in predictive value provided by the 15 most important predictors included in the ambulance data-based models is reported in Fig 2, in order of descending mean gain across the 3 outcomes. Patient age and the provision of oxygen (coded as the liter per minute flow) ranked highest, followed by a number of patient vital signs. Whether or not the patient was transported using lights and sirens to the hospital was a strong predictor of outcomes. A number of measures of call duration (time to the hospital, time on-scene, and time between call receipt and ambulance dispatch), the distance to the nearest ED, and time of day of the call also ranked highly. A summary of the gain provided by all included variables is provided in S1 Table.

**Table 4. Outcomes by priority compared with hypothetical dispatch prioritization.**

| Type | Priority | N | Emergent transport, percent | Ambulance intervention, percent | NEWS value, mean | Admitted, percent | Critical care, percent | 2-day mortality, percent |
|------|----------|---|------------------------------|----------------------------------|-------------------|-------------------|-------------------------|---------------------------|
| Current | 1A | 473 | 35.3 (31.3–39.7) | 87.5 (84.6–90.5) | 5.64 (5.28–6.02) | 48.6 (44.2–52.9) | 14.6 (11.6–17.8) | 4.7 (3.0–6.6) |
| | 1B | 5657 | 22.5 (21.4–23.6) | 86.4 (85.5–87.2) | 3.70 (3.59–3.80) | 52.3 (51.0–53.6) | 6.5 (5.9–7.2) | 1.7 (1.4–2.0) |
| | 2A | 6112 | 3.4 (3.0–3.9) | 70.5 (69.3–71.6) | 2.81 (2.74–2.89) | 51.5 (50.1–52.8) | 4.7 (4.2–5.2) | 0.6 (0.4–0.9) |
| | 2B | 1353 | 1.7 (1.1–2.4) | 60.5 (57.9–63.2) | 2.33 (2.20–2.46) | 47.7 (44.8–50.5) | 4.4 (3.3–5.4) | 0.4 (0.1–0.9) |
| Hypo-thetical | 1A | 473 | 37.8 (33.4–42.5) | 90.7 (87.9–93.2) | 7.22 (6.84–7.61) | 75.3 (71.2–79.1) | 22.2 (18.4–25.8) | 10.1 (7.4–12.9) |
| | 1B | 5657 | 15.3 (14.3–16.2) | 77.9 (76.8–79.0) | 4.22 (4.13–4.33) | 69.6 (68.5–70.8) | 8.2 (7.5–8.9) | 1.6 (1.2–1.9) |
| | 2A | 6112 | 8.9 (8.1–9.7) | 74.9 (73.8–76.0) | 2.37 (2.31–2.44) | 40.3 (39.0–41.5) | 3.3 (2.8–3.7) | 0.4 (0.3–0.6) |
| | 2B | 1353 | 6.1 (4.9–7.3) | 75.0 (72.7–77.2) | 1.55 (1.45–1.65) | 16.3 (14.3–18.4) | 1.0 (0.5–1.6) | 0.0 (0.0–0.0) |

Table presents a comparison of outcome prevalences within each priority group as dispatched per current clinical practice, and for a hypothetical situation in which calls were dispatched based solely on the proposed dispatch risk score. All estimates are reported with bootstrapped 95% confidence intervals.

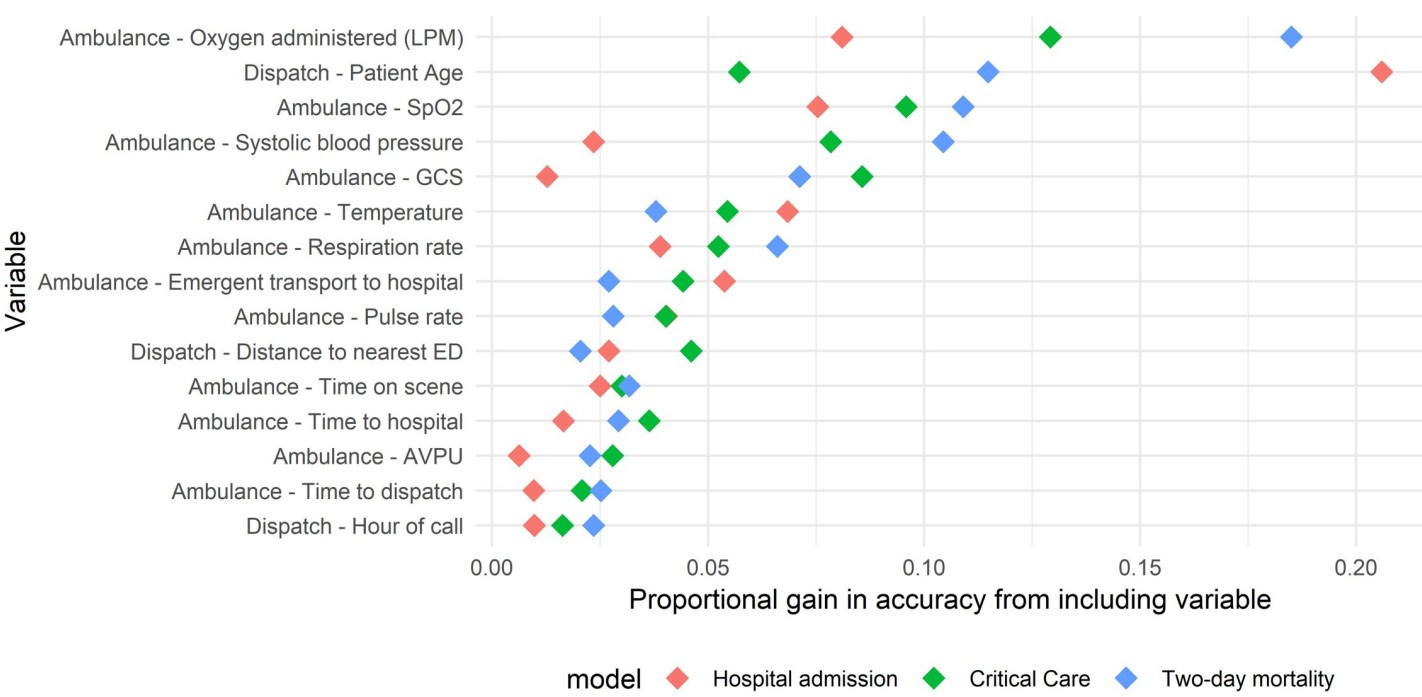

**Fig 2. Importance of variables in predicting hospital outcomes in Ambulance models.** Variables are arranged in order of descending mean gain across the models predicting the outcomes included in the ambulance data-based risk score.

## Discussion

### Limitations

We limited this study to the investigation of a composite score based on an unweighted average of model predictions for three specific hospital outcomes. In doing so, we make the assumption that each of these outcomes is equally important in determining the overall risks associated with the patient. A sensitivity analysis provided in S3 Table demonstrated that while the predictive value of the risk scores did shift in favor of more heavily weighted outcomes across a range of weights, the differences did not impact the main findings of this study. The unweighted average furthermore offered a good compromise in terms of discrimination for each of the constituent outcomes. The most appropriate set of outcomes and associated weights to employ is nevertheless dependent on the intended application of the risk scores, and we recognize that we have examined only one of many potentially valid sets of outcome measures to employ in prehospital risk assessment.

We employed a relatively restrictive set of inclusion criteria in this study which excluded patients left at the scene of the incident, and patients transported to non-ED destinations. We considered investigation of the cohort of patients transported to an ED to provide the most replicable results as this avoided several sources of loss to follow-up present among non-transported patients. Given that prior research on ML-based risk scores has been performed primarily in the context of the ED, this cohort was thought to be most relevant to other researchers. We performed a sensitivity analysis to investigate the impact of using a broader patient cohort including patients left on scene by the ambulance and those transported to destinations other than the ED. The results were essentially unchanged as reported in S2 Analysis. Upon implementing these methods, care must be taken to ensure that the criteria used to include patients in a training dataset results in a population of patients similar to those upon whom the risk assessment tools will be applied.

We observed a rate of loss to follow up of around 5–10% upon the application of each of our exclusion criteria. To assess and ameliorate risks associated with data quality issues, we manually spot-checked records to ensure the accuracy of our automated data extraction methods, finding an average accuracy of 96% for the ambulance and hospital measures. We addressed systematic data extraction issues where we found them, which could account for the lower rate of loss to follow-up we observed in the test dataset as reported in Table 1. The linkage rates found in this study were similar or superior to other studies of prehospital data [45–47]. We also observed c-index values for NEWS scores similar to those found in previous studies; Lane et al. [18] identified c-indexes of 0.85 for NEWS in predicting two-day mortality, similar to our value of 0.85 (0.82–0.86). Results were also similar to those identified by Pirneskoski et al. [19], who found a c-index value of 0.84 for NEWS scores in predicting 1-day mortality. Such agreement suggests that the quality of the data in this study is comparable to that of previously published research in the field.

While the ML models reported on in this single-site study performed well in prospective validation, they are not likely to generalize well if applied directly to other contexts. Guidelines regarding hospital admission and intensive care for instance may vary, potentially biasing outcome predictions if these models were applied directly in other settings. Such idiosyncrasies are likely to exist among predictor variables as well: Oxygen was found to have been administered to 17% of patients in this study for instance, a rate which appears to be lower than that found in other contexts [48,49]. In settings where oxygen is administered more liberally, it is not likely to be as strongly associated with patient acuity. The ML framework we employ is however highly flexible, and is likely to produce good results if models were to be trained "from scratch" on other similar datasets. As such, rather than seek to apply the specific models developed in this study to other settings, we encourage researchers to generate and validate novel models based on the framework we propose in other settings. To enhance reproducibility, we sought to adhere to TRIPOD guidelines in reporting our results regarding the development and validation of these models [50], and it is hoped that the source code found in S1 Code will facilitate the replication of-, and improvement upon our results.

## Interpretation

Overall, these findings suggest that the application of machine learning methods using routinely collected dispatch and ambulance data is a feasible approach to improving the decision support tools used by prehospital care providers to assess patients in terms of the need for hospital care. We found that risk scores generated using ML models based on ambulance data outperformed NEWS in predicting hospital outcomes. Risk scores based on data gathered at the EMD center outperformed the prioritizations made by dispatch nurses, and performed comparably to NEWS (which are based on physiological data gathered upon patient contact) in settings where high sensitivity is demanded. Model performance was similar when validated internally using cross-validation and when evaluated in a prospectively gathered dataset, suggesting that the performance of the models is likely to remain stable upon being implemented within the studied context. ML-based risk scores demonstrated acceptable levels of calibration both overall and stratified by age, gender, priority and common call types, and were only mildly sensitive to the selection of alternate sets of weights.

During the development of the methods reported here, we investigated the performance of a number of ML techniques including regularized logistic regression, support vector machines, random forests, gradient boosting, and deep neural networks in the training dataset. As in previous studies [27,51–53], we found that the XGBoost algorithm performed at least as well as other methods we applied to these data in terms of discrimination. We also found that the

XGBoost algorithm had several practical benefits, including being essentially invariant to monotonic transformations of the predictors, thus simplifying the data transformation pipeline [40], and appropriately handling missing data using a sparsity-aware splitting algorithm [39]. While providing good discrimination, the approach does have some drawbacks including being somewhat difficult to interpret, the inability to update models without access to the full original dataset, and that the models are not inherently well calibrated as logistic regression for instance is. We employ a novel method for combining the predictions of these models into a composite measure using a weighted average. This provides some practical benefits; It allows the instrument to be tailored to various applications by adjusting the weights associated with each prediction, and it allows the presentation of predictions for each individual component outcome to users, in addition to an overall risk score which can be rapidly interpreted. This approach also avoids the information loss associated with binning continuous values as employed in previous studies [26,32], which is particularly undesirable in applications where the acuity of patients is to be directly compared.

We found the overall calibration of our composite risk scores to be satisfactory, despite their nature as an average of multiple outcomes. Examination of calibration across sub-populations yielded interesting results which could be further examined. We found NEWS for instance to systematically under-estimate the probability of hospital admission among older patients—Such miscalibration could be the result of an over-estimation of risks among older patients in the hospital admission process, but could also represent an underlying bias in NEWS as currently calculated. Interestingly, all risk scores tended to underestimate the probability of two-day mortality for the oldest quartile of patients. While the usual caution in interpreting post-hoc sub-group analyses is warranted, we found analyses of this type to be useful in developing the models reported here, and in considering how to proceed with their application to clinical practice.

While dispatcher prioritizations did have some predictive value for critical care and two-day mortality, their discrimination was poor in comparison with all other risk assessment instruments with regards to hospital outcomes. This may in part be due to dispatchers prioritizing ambulance responses with an eye to the need for prehospital rather than in-hospital care. These aspects of patient care often coincide but can in some cases differ. Cases of severe allergic reactions for instance call for a high priority ambulance response, but following treatment in the field by ambulance staff, often require only minimal in-hospital care. This is reflected in the results reported in Table 4, which demonstrated that while the risk scores improved differentiation with regards to hospital outcomes, it did not improve the stratification of patients with regards to prehospital interventions. These findings highlight the need to use this type of risk assessment instrument in the context of clinical guidelines which permit care providers to override the risk score in complex situations where factors beyond those captured in the models are at play. We suggest that the results of such an instrument should rather be considered as one factor in a holistic patient assessment.

Our models generally had lower levels of overall predictive value than found in previous studies investigating these outcomes based on data collected at the ED. This could in part be explained by population differences, given that the population of ambulance-transported patients investigated here constitutes a sum-population of the highest-acuity patients cared for at the ED [54–56]. The population in this study for instance had an average rate of in-hospital mortality of 4%, compared with the 0.5% rate found by Levin et al. [26], while our hospital admission rate was 52% as compared with the 30% found by Hong et al. [27], both of whom studied the full population of ED patients. We provide outcome data stratified by patient condition at the ED as S4 Table to allow comparison with other studies. It is also the case that the data available in records of prehospital care tend to be less detailed, lacking granular

information regarding for instance the patient's past medical history and laboratory test results. Such data have been found to provide substantial improvements to patient outcome predictions [27,29]. This study demonstrates that despite these barriers, prehospital data does have value in predicting hospital outcomes. We identified no studies of ED triage models which included prehospital data, and as such, we suggest that one avenue for improving the performance of in-hospital triage models may be to include variables drawn from dispatch and ambulance records.

The risk scores we present in this study have potential applications throughout the chain of prehospital emergency care including at the dispatch center, on the ambulance, and at the ED. While each potential use case will require the models to be tailored to suit the application, it is a strength of the approach we propose that such adaptations are relatively simple to implement. At the dispatch center for instance, the risk scores could be applied directly for the task of assessing the need for hospital care. As noted previously, different outcome measures must be selected to capture the need for prehospital care however. While a number of measures have been proposed to assess dispatch accuracy, more research is needed to establish consensus regarding the most suitable set of measures to employ [57]. In the ambulance setting, the risk scores appear suitable for use as a direct replacement for NEWS for the identification of patients for whom referral to a hospital is necessary [58]. In this application, broader inclusion criteria such as those reported in S2 Analysis should be applied in order to train models on the appropriate cohort of patients.

The inclusion criteria used in the main analysis are most applicable to the use case of alerting receiving EDs to incoming high-risk patients. For instance, it may be possible to shorten ED waiting times by identifying incoming patients with a high probability of admission. Staff could then make arrangements for in-patient care immediately upon patient hand off, or perhaps even before the patient arrives at the ED. In this application, the composite measures we propose have similar levels of discrimination with regards to hospital admission as single models dedicated to predicting only this outcome, while retaining a high predictive value for other outcomes of interest as reported in S3 Table. To demonstrate the behavior of the risk scores we present, a simplified version of the ambulance data-based risk assessment instrument has been made publicly available and integrated into a web application for non-clinical use [59].

In conclusion, these results demonstrate that machine learning offers a viable approach to improving the accuracy of prehospital risk assessments, both in relation to existing rule-based triage algorithms, and current practice. Further research should investigate if the inclusion of additional unstructured data such as free-text notes and dispatch center call recordings could further improve the predictive value of the models reported here. Studies to investigate the attitudes of care providers with regards to risk assessments using ML may also prove fruitful; while ML methods can provide prehospital care providers with a more accurate risk score, the lack of direct interpretability often associated with such models may prove to be a barrier to acceptance. This study establishes only the feasibility of this approach to prehospital risk assessment, and further studies must establish the ability of this approach to influence the decisions of care providers and impact patient outcomes by means of prospective, preferably randomized, trial.

## Supporting information

**S1 Analysis. Precision/Recall analysis.** Provides results from a Precision/recall curve analysis as commonly reported in the machine learning literature, presented in the same manner as Fig 1 and Table 2 in the main analysis.
(DOCX)

**S2 Analysis. Broad inclusion criteria.** Investigates the sensitivity of our results to a broadening of the inclusion criteria used in the main analysis, we report all tables, figures and supplementary analyses presented here, but including patients left on scene by an ambulance or transported to a non-ED destination.
(DOCX)

**S1 Fig. Model calibration curves.** Provides the results of model calibration analyses using lowess smoothed calibration curves for both overall calibration, and calibration among sub-populations divided by age quartile, gender, call priority, and the 5 most common call types.
(DOCX)

**S1 Table. Predictor description.** Descriptions of each set of predictors included in gradient boosting models, providing information regarding the number of non-missing, non-zero values among included calls, the average gain provided by the predictor, and the number of dummy-encoded variables included from the predictor in the models.
(DOCX)

**S2 Table. Model calibration mean average error.** Provides summary statistics in the form of the mean average calibration error for NEWS and ML risk scores both in the full population, and the weighted average of all investigated sub-populations.
(DOCX)

**S3 Table. Sensitivity to alternate weights.** Reports c-indexes for risk scores across a range of alternate weighting schemes, including the performance of individual model predictions across all investigated outcomes.
(DOCX)

**S4 Table. Descriptive statistics by ED triage category.** Presents patient volumes, ages, and outcomes across ED triage categories with more than 300 occurrences.
(DOCX)

**S1 Code. R Source code.** Provides all R code necessary to replicate the results reported in this manuscript in a user-provided dataset. If no dataset is provided, results are calculated in a randomly generated synthetic dataset mimicking the univariate properties of our data. This repository also includes the public release models and the code for an interactive demonstration of the models using the Shiny wed app framework. A maintained version of this code may be found on github: https://github.com/dnspangler/openTriage_validation, and an interactive demo app based on the public release models may be found here: https://ucpr.se/openTriage_demo/.
(ZIP)

## Author Contributions

**Conceptualization:** Douglas Spangler, Hans Blomberg.

**Data curation:** Douglas Spangler, Thomas Hermansson.

**Formal analysis:** Douglas Spangler.

**Funding acquisition:** Hans Blomberg.

**Investigation:** Douglas Spangler, Thomas Hermansson, David Smekal.

**Methodology:** Douglas Spangler, Hans Blomberg.

**Project administration:** Douglas Spangler.

**Software:** Douglas Spangler.

**Supervision:** Hans Blomberg.

**Validation:** Thomas Hermansson.

**Writing – original draft:** Douglas Spangler.

**Writing – review & editing:** Douglas Spangler, Thomas Hermansson, David Smekal, Hans Blomberg.

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
