## [Decision Letter · Decision Letter 0]

23 Oct 2019

PONE-D-19-25317

A validation of machine learning-based risk scores in the prehospital setting

PLOS ONE

Dear Mr. Spangler,

Thank you for submitting your manuscript to PLOS ONE. After careful consideration, we feel that it has merit but does not fully meet PLOS ONE’s publication criteria as it currently stands. Therefore, we invite you to submit a revised version of the manuscript that addresses the points raised during the review process.

We would appreciate receiving your revised manuscript by Dec 07 2019 11:59PM. To enhance the reproducibility of your results, we recommend that if applicable you deposit your laboratory protocols in protocols.io, where a protocol can be assigned its own identifier (DOI) such that it can be cited independently in the future. For instructions see: http://journals.plos.org/plosone/s/submission-guidelines#loc-laboratory-protocols

We look forward to receiving your revised manuscript.

Kind regards,

Itamar Ashkenazi

Academic Editor

PLOS ONE

Journal Requirements:

2. In the ethics statement in the manuscript and in the online submission form, please provide additional information about the patient records used in your retrospective study. Specifically, please ensure that you have discussed whether all data were fully anonymized before you accessed them and/or whether the IRB or ethics committee waived the requirement for informed consent. If patients provided informed written consent to have data from their medical records used in research, please include this information.

Additional Editor Comments (if provided):

Dear Author,

Your article has been evaluated. In this letter you will find two sets of comments, by the reviewer and myself. We both found value in your study. We both think however, there is a need to emphasize certain issues.

The introduction needs to better define the problem being studied: Is it the need to better identify high risk patients who may suffer adverse outcome (ICU, death)? Is it to identify those who do not need hospitalization? Is it both? Please correct me if I am wrong, but the population and endpoints defined for this study do not allow the study of the identification of patients who do not need hospitalization. First, many such patients may be have been excluded as stated by the exclusion criteria. Second, the end points evaluated were not focused on this population.

If the emphasis is to identify those with adverse outcome (ICU, death), why is the composite score of any importance?

The emphasis in the results is on ROC curves, which is fair... However, as clinicians who work in this environment everyday, we cannot disregard the actual endpoints if the machine learning technique is implemented. In other words, based on the results obtained in Uppsala, if a certain ("optimal") machine learning score is decided upon, how much better will it triage patients compared to what we see in table 1. Take for example the baseline of 1.6-1.7% ICU admissions, 3.7-3.9% hospital deaths in 2A and 2B categories (which are less than optimal).

Sincerely,

Itamar Ashkenazi

Reviewers' comments:

Reviewer's Responses to Questions

**Comments to the Author**

1. Is the manuscript technically sound, and do the data support the conclusions?

Reviewer #1: Partly

2. Has the statistical analysis been performed appropriately and rigorously? 

Reviewer #1: Yes

3. Have the authors made all data underlying the findings in their manuscript fully available?

Reviewer #1: Yes

4. Is the manuscript presented in an intelligible fashion and written in standard English?

Reviewer #1: Yes

5. Review Comments to the Author

Reviewer #1: 1. This is hard for this reviewer to understand whether this model is going to be used at dispatch center, on the ambulance, or at emergency departments. It is clear that there are differences in triage setting between these units depending on the fact that a good triage is made of what you see (Anatomy), what you get (Vital signs) and what you perceive (clinical judgement). There is a lack of discussion about where such ML-based risk assessment/triage should be used and a comparison of where it can be more valuable/suitable.

2. Is this model a risk assessment model or a triage model? A triage model should be Fair, quick, feasible and have continuity along the chain of response. This model may be quick and feasible, but can hardly be fair since triage is not only scores and risks, it is also clinical judgement and experience. The latter can be learned, but the former remains in hand of performer. Another major point is the lack of continuity. How can it be guaranteed that the triage model is sustainable all the way?

3. You use risk factors to build up a bank of knowledge by which you get predictors and can prioritize different conditions. We know that risks may change due time. How can you make sure that the machine is always updated? Yearly calibration is not enough? Furthermore, higher risk patients should not necessarily be prioritized first when there might be other situations that should be handled first?

4. How accurate and well-analyzed is your data? E.g., on page 19 line 335 “Oxygen was found to have been administered to 17% of patients in this study for instance, a rate which appears to be lower than that found in other contexts. We know that almost all patients receives oxygen during transportation or at ED, but this is not regularly noted in their medical files.

5. There is no information given on diagnoses included in this study. The outcome of this study may change depending on the diagnoses. Benign diagnoses, easier to assess by machine.

6. EMS faces many challenges not only aging people but also the increased severity of traumas, complexity of patients, and unnecessary use of EMS (public education). How can this model influence these parameters?

7. There are other papers describing this model. What does your study add to what we already know?

6. PLOS authors have the option to publish the peer review history of their article (what does this mean?). If published, this will include your full peer review and any attached files.

Reviewer #1: No

---

## [Author Response · Author response to Decision Letter 0]

19 Nov 2019

We thank the editor and reviewer for their valuable comments and suggestions for improvement regarding our manuscript. We hope that you will find our revised manuscript suitable for publication in PLoS One.

We agree that our motivation for using the composite score and our discussion of potential applications in particular required substantial clarifications. Please find below our response to each of the comments in turn. Major revisions and additions noted in our response have also been highlighted yellow in the manuscript. 

We have now also developed a simplified version of the ambulance data-based instrument suitable for public release, and implemented it in an interactive web application. Our hope is that this will enable the reviewers (and your readership) to explore the behavior of our models and to gain an intuition about how they might be employed (referenced in the manuscript on p. 27, lines 510-512):

https://ucpr.se/openTriage_demo

Please note that the source code for this tool and a document reporting results for the public release models are included in S8 Code. We thank the editor and the reviewer for their time and for their contributions to improving the manuscript.

Sincerely,

Douglas Spangler, Corresponding Author

Editor:

The introduction needs to better define the problem being studied: Is it the need to better identify high risk patients who may suffer adverse outcome (ICU, death)? Is it to identify those who do not need hospitalization? Is it both? Please correct me if I am wrong, but the population and endpoints defined for this study do not allow the study of the identification of patients who do not need hospitalization. First, many such patients may be have been excluded as stated by the exclusion criteria. Second, the end points evaluated were not focused on this population.

If the emphasis is to identify those with adverse outcome (ICU, death), why is the composite score of any importance?

Author response:

We agree and have added substantially to the discussion of our goals and motivations for using a composite score. It is our aim to develop an instrument to predict the risk of patient deterioration, similar to how the National Early Warning Score (NEWS) is commonly used in prehospital care today. We have clarified this on p. 4, lines 64-68. As “patient deterioration” cannot be measured directly, we selected these three endpoints as measures to operationalize this concept based on previous literature validating NEWS and other ML-based ED risk scores. We have included additional motivation for using a composite score to capture the combined likelihoods of these outcomes on p. 5, lines 77-92. We have also included some further discussion of the implications of our proposed approach on p. 24, lines 435-443.

We have added additional reasoning surrounding our choice of inclusion criteria on p. 21, lines 365-374. In our quality improvement work, we do indeed often utilize broader patient cohorts. We have now included a sensitivity analysis (S6 Analysis) reporting all of the tables and figures presented in the manuscript, but applied to the broadest cohort of patients within which valid comparisons with NEWS can still be made. The results in this sensitivity analysis were remarkably similar to those reported in our main analysis.

The emphasis in the results is on ROC curves, which is fair... However, as clinicians who work in this environment everyday, we cannot disregard the actual endpoints if the machine learning technique is implemented. In other words, based on the results obtained in Uppsala, if a certain ("optimal") machine learning score is decided upon, how much better will it triage patients compared to what we see in table 1. Take for example the baseline of 1.6-1.7% ICU admissions, 3.7-3.9% hospital deaths in 2A and 2B categories (which are less than optimal).

Author response:

This is an excellent suggestion for conveying the potential impact of using the risk scores. We have included a Table 4 which reports the hypothetical distribution of outcomes across priority groups if calls had been prioritized at the dispatch center strictly per the dispatch risk scores (Noted in the manuscript under methods: p. 12, lines 230-233, results: p. 18, lines 323-336, and discussion: p. 25, lines 462-469). As suggested by the findings of our study and by the reviewer below however, such a strict use of these scores would not actually be appropriate in practice. 

Reviewer #1: 

1. This is hard for this reviewer to understand whether this model is going to be used at dispatch center, on the ambulance, or at emergency departments. It is clear that there are differences in triage setting between these units depending on the fact that a good triage is made of what you see (Anatomy), what you get (Vital signs) and what you perceive (clinical judgement). There is a lack of discussion about where such ML-based risk assessment/triage should be used and a comparison of where it can be more valuable/suitable.

Author response:

We agree that there is a lack of discussion regarding the applications of the risk scores we present. Our intent was to limit the scope of our manuscript to a statistical validation of our methods, but we see now that some discussion of applications is necessary to provide context. We are planning interventions in each of the domains the reviewer mentions. As the reviewer suggests, each of these applications have unique considerations which can only be explored fully in their own separate studies. 

While we hope to preserve the focus of the manuscript on the statistical validation of our proposed methods, we have substantially deepened our discussion of the potential applications of the risk scores (p. 6, lines 93-103 and p. 26, lines 488-509). We see it as a strength of the study that the methods are validated in a variety of prehospital contexts. 

2. Is this model a risk assessment model or a triage model? A triage model should be Fair, quick, feasible and have continuity along the chain of response. This model may be quick and feasible, but can hardly be fair since triage is not only scores and risks, it is also clinical judgement and experience. The latter can be learned, but the former remains in hand of performer. Another major point is the lack of continuity. How can it be guaranteed that the triage model is sustainable all the way?

Author response:

We consider this to be a risk assessment instrument. As with any such instrument it must be used within the context of a comprehensive triage system, and we too believe that they should serve as a compliment to the clinical expertise of the care provider as we now explicate on p. 25 lines 465-469. We kindly ask for clarification regarding the questions of continuity and being “sustainable all the way” if the reviewer feels that they have not been addressed in our response.

3. You use risk factors to build up a bank of knowledge by which you get predictors and can prioritize different conditions. We know that risks may change due time. How can you make sure that the machine is always updated? Yearly calibration is not enough? Furthermore, higher risk patients should not necessarily be prioritized first when there might be other situations that should be handled first?

Author response:

We plan to update and evaluate models weekly upon implementation into clinical practice, which is the frequency at which we receive updated outcome data from the hospitals. In consideration of this comment, we have updated the definition of the “critical care” outcome to include only in-hospital deaths within 30 days of ED contact (noted on p. 9, lines 165-167). The change affected 130 observations (4% of the 3018 previous instances of critical care) and had no notable impact on our findings. This change allows for valid updated models to be generated based on outcome data with a 30-day lag.

We agree that keeping human care providers “in the loop” is critical for identifying situations in which a clinical risk score might not be appropriate to employ due to the social or logistical complexities of a given situation. As we now note on p. 25, lines 465-469, we encourage the use of guidelines permitting risk scores to be overridden at the discretion of care providers.

4. How accurate and well-analyzed is your data? E.g., on page 19 line 335 “Oxygen was found to have been administered to 17% of patients in this study for instance, a rate which appears to be lower than that found in other contexts. We know that almost all patients receives oxygen during transportation or at ED, but this is not regularly noted in their medical files.

Author response:

We performed random spot checks (using oversampling for rare outcomes) to assess the quality of the data used in this study, finding an average of 96% accuracy in data extracted from ambulance and hospital records. We have noted this figure in the manuscript (p. 21, line 381-382). The 299 records which we investigated for the presence of oxygen administration specifically had an accuracy of 100% as compared with a manual review of full ambulance and hospital journals. One explanation for the low rate of oxygen administration is that in Sweden, the value of oxygen therapy has been highly questioned in recent years, and our clinical guidelines state that it is not to be provided unless necessary based on a low SpO2 reading. Two previous studies reporting the prevalence of oxygen administration within cohorts of unselected ambulance patients found oxygen administration rates of 47% and 34% respectively. 

5. There is no information given on diagnoses included in this study. The outcome of this study may change depending on the diagnoses. Benign diagnoses, easier to assess by machine.

Author response:

We agree that this is a gap in the descriptive data we provide. We now include additional descriptive statistics and outcomes stratified by the most common symptoms documented by ED staff as appendix S7 Table and reference this in the manuscript on p. 25, lines 477-479. We also agree that benign diagnoses are easier for models to spot – We cite this as one of the potential reasons for the lower levels of discrimination found in this study as compared with studies using similar models applied to cohorts of ED patients which include lower-risk walk-in patients.

6. EMS faces many challenges not only aging people but also the increased severity of traumas, complexity of patients, and unnecessary use of EMS (public education). How can this model influence these parameters?

Author response:

The reasoning above is why we believe that new risk assessment instruments are needed in prehospital care. Simple triage algorithms such as NEWS do not take into account risk factors including the patient’s past medical history, current signs and symptoms, interventions performed by the caregiver, or even patient demographics. As such, their power to differentiate between the increasingly complex and multifaceted patient conditions presenting to prehospital care systems is limited. As noted above, we have added additional discussion of potential applications which could influence how we are able to navigate patients through the prehospital care system using these tools on p. 6, lines 93-103 and p. 26, lines 488-509.

7. There are other papers describing this model. What does your study add to what we already know?

Author response:

The main novelty of this study is twofold. Firstly, this study extends the use of machine learning methods to the field of prehospital care. We are unaware of any previous studies investigating similar approaches, and this study establishes feasibility and a first benchmark for model performance in the prehospital context (noted on p. 23, lines 410-413). We hope that the R code which we provide will be useful to other researchers seeking to build upon the methods we report. Secondly, we believe that the method of using a weighted average of model predictions to generate a composite risk scores is a novel approach to aggregating multiple outcome measures into a simple risk score which may be rapidly interpreted by care providers (noted on p. 24, lines 435-443).

---

## [Decision Letter · Decision Letter 1]

2 Dec 2019

A validation of machine learning-based risk scores in the prehospital setting

PONE-D-19-25317R1

Dear Dr. Spangler,

We are pleased to inform you that your manuscript has been judged scientifically suitable for publication and will be formally accepted for publication once it complies with all outstanding technical requirements.

With kind regards,

Itamar Ashkenazi

Academic Editor

PLOS ONE

Additional Editor Comments (optional):

Reviewers' comments:

Reviewer's Responses to Questions

**Comments to the Author**

1. If the authors have adequately addressed your comments raised in a previous round of review and you feel that this manuscript is now acceptable for publication, you may indicate that here to bypass the “Comments to the Author” section, enter your conflict of interest statement in the “Confidential to Editor” section, and submit your "Accept" recommendation.

Reviewer #1: All comments have been addressed

2. Is the manuscript technically sound, and do the data support the conclusions?

Reviewer #1: Yes

3. Has the statistical analysis been performed appropriately and rigorously? 

Reviewer #1: Yes

4. Have the authors made all data underlying the findings in their manuscript fully available?

Reviewer #1: Yes

5. Is the manuscript presented in an intelligible fashion and written in standard English?

Reviewer #1: Yes

6. Review Comments to the Author

Reviewer #1: (No Response)

7. PLOS authors have the option to publish the peer review history of their article (what does this mean?). If published, this will include your full peer review and any attached files.

Reviewer #1: Yes: Amir Khorram-Manesh

---

## [Editor Report · Acceptance letter]

5 Dec 2019

PONE-D-19-25317R1 

A validation of machine learning-based risk scores in the prehospital setting 

Dear Dr. Spangler:

I am pleased to inform you that your manuscript has been deemed suitable for publication in PLOS ONE. Congratulations! Your manuscript is now with our production department. 

With kind regards,

on behalf of

Dr. Itamar Ashkenazi 

Academic Editor

PLOS ONE